# Advancing Medical Image Segmentation with Self-Supervised Learning: A 3D Student-Teacher Approach for Cardiac and Neurological Imaging

**Moona Mazher**[*,1] (ID)                  M.MAZHER@UCL.AC.UK
**Daniel C. Alexander**[1]                 D.ALEXANDER@UCL.AC.UK
**Abdul Qayyum**[2,3]                 A.QAYYUM@IMPERIAL.AC.UK
**Steven A Niederer**[2,3]              S.NIEDERER@IMPERIAL.AC.UK

[1] *Hawkes Institute, Department of Computer Science, University College London, London, United Kingdom*

[2] *National Heart and Lung Institute, Faculty of Medicine, Imperial College London, London, United Kingdom*

[3] *The Alan Turing Institute, London, United Kingdom*

**Editors:** Accepted for publication at MIDL 2025

## Abstract

We propose 3D-SegSync, a self-supervised learning (SSL) framework designed to improve segmentation accuracy for both cardiac and neurological structures. It integrates a student-teacher model with a 3D Vision-LSTM (xLSTM) backbone to capture spatial dependencies in volumetric data. The SSL phase utilizes large-scale unlabeled datasets for pretraining, followed by fine-tuning on labeled data to improve segmentation across CT and MRI scans. Experimental results demonstrate that 3D-SegSync achieves consistent performance across different anatomical structures. Additionally, its ability to generalize between CT and MRI without requiring modality-specific modifications highlights its adaptability for cardiac and neurological image segmentation. Given its strong performance, 3D-SegSync has the potential to be extended to other medical image segmentation tasks in the future. Code can be found here: https://github.com/Moona-Mazher/3D-SegSync_SSL.

**Keywords:** Self-Supervised Learning (SSL), Whole Heart Segmentation (WHS), Ischemic Stroke Lesion Segmentation (ISLES), CT Imaging, MRI Imaging, Cardiac Imaging, Neurological Imaging, xLSTM, Multi-Modal Imaging, Traumatic Brain Injury (TBI).

## 1. Introduction

Medical image segmentation is critical for accurate diagnosis, treatment planning, and monitoring disease progression, especially in complex 3D tasks such as cardiac and neurological imaging. However, segmentation in these areas remains challenging due to factors like limited annotated data, modality variability, and suboptimal image quality. These difficulties are particularly evident in cardiac and brain imaging, where anatomical complexity, patient variability, and motion artifacts add complexity.

**Challenges in Cardiac and Neurological Imaging:** In cardiac imaging, accurate segmentation of structures like ventricles, atria, myocardium, and blood vessels is essential

---

[*] Corresponding Author

for diagnosing heart disease. However, the dynamic shape changes across the cardiac cycle, modality variability (CT vs. MRI), and motion artifacts make segmentation difficult. Existing methods, such as those by (Zhuang and Shen, 2016) and (Isensee et al., 2019), often struggle with multi-center datasets and modality generalization.

In neurological imaging, accurate segmentation of ischemic stroke lesions is critical for prognosis and treatment. While MRI is commonly used, the variability in brain anatomy, lesion complexity, and imaging artifacts present substantial challenges. Models like (Menze et al., 2015) and (Kohl et al., 2020) have demonstrated robust performance but tend to rely on large annotated datasets and struggle with generalization across different clinical settings and modalities.

**Gaps in Existing Approaches:** (1) SSL methods like SimCLR (Chen et al., 2020) and MoCo (He et al., 2020) have been successful in 2D tasks but fail to capture the long-range spatial dependencies and complex volumetric data of 3D medical images. Recent advancements have introduced 3D SSL models such as SwinMM (Wang et al., 2023), SwinSSL (Tang et al., 2022), VoCo (Wu et al., 2024), and Hi-End-MAE (Tang et al., 2025) to handle volumetric data of 3D medical images. However, these models still face limitations in fully adapting to the 3D nature of medical data, particularly in accurately modeling the intricate spatial relationships and improving segmentation performance across the entire volume of the image. These methods often focus on learning low-level features from local patches, but they may not explicitly model long-range dependencies between slices, which is crucial for accurate segmentation in 3D data. (2) Modality-Specific Limitations: Many existing models are optimized for specific imaging modalities (CT or MRI) and struggle to generalize across different modalities, leading to reduced performance in multi-modal settings (Ronneberger et al., 2015a); (Zhu et al., 2021). (3) Dependence on Labeled Data: Despite the promise of SSL, most methods still require substantial labeled datasets for fine-tuning, which remains a bottleneck in medical imaging due to the cost and time involved in manual annotation.

**Contribution:** Our study integrates 3D self-supervised learning (SSL) pretraining with xLSTM for medical image segmentation, particularly for large-scale CT and MRI datasets in cardiology and neurology. Inspired by the DINOv2 student-teacher framework, we extend its principles from 2D to 3D by replacing Vision Transformers (ViTs) with a 3D xLSTM-based encoder. While state-of-the-art SSL models like DINOv2, MAE, SwinMM, and SwinSSL rely on ViTs for feature learning, our approach leverages xLSTM to capture long-range spatial dependencies across slices, making it more effective for volumetric medical imaging. Pretrained on large unlabeled datasets and fine-tuned on smaller labeled ones, our model enhances segmentation accuracy and reliability in cardiology and neurology by leveraging SSL and xLSTM for improved feature representation and anatomical structure learning.

Our SSL-based framework helps overcome key challenges in cardiac and neurological imaging, including limited labeled data and cross-modality segmentation. By improving feature learning and spatial dependency modeling, our approach enhances segmentation performance and adaptability. This work has the potential to support clinical decision-making and improve patient outcomes in the future.

## 2. Proposed Method

### 2.1. Dataset

We curated and preprocessed whole heart CT/MRI and brain MRI datasets for self-supervised learning (SSL) and segmentation tasks. For whole heart segmentation, we used CT Coronary Angiography (CTCA) (Gharleghi et al., 2022) from the Coronary Atlas, ImageCAS (1,000 patients) (Zeng et al., 2023), ImageTBAD (56 CT angiography images) (Radl et al., 2022), and TotalSegmentator (1,204 CT scans) (Wasserthal et al., 2023). Unlabeled validation datasets (held by the challenge organizers to evaluate the participating teams' performance) from the MMWHS (Zhuang et al., 2019) and WHS++ (Zhuang and Shen, 2016) challenges were included for SSL pretraining, while labeled training sets from MMWHS, WHS++, and HVSMR-2.0 (Pace et al., 2024) were used for fine-tuning. For brain imaging SSL pretraining, we leveraged ISLES datasets, including ISLES 2022 (400 MRI cases) (de la Rosa et al., 2024) and previous versions (ISLES 2015, 2016, 2018), along with ATLAS (Liew et al., 2022) (304 cases in v1.2, 1,271 in v2.0). The model was fine-tuned on ISLES 2024 for stroke lesion segmentation and Traumatic Brain Injury (TBI) leision segmentation. The dataset distribution followed three phases: 1. pretraining on large, unlabeled (Cardiac: CT/MRI) (Brain: MRI) datasets for general feature learning, 2. fine-tuning with labeled datasets for the heart (HVSMR-2.0, MMWHS-CT, WHS++CT, MMWHS-MRI, WHS++MRI) and brain (ISLES 2024, TBI) segmentation These datasets were split into 80% for training and 20% for testing. 3. Finally, in the testing phase, we evaluated the model on the remaining 20% of labeled data for both heart and brain segmentation tasks to assess its performance after pretraining and fine-tuning.

### 2.2. Proposed Framework for SSL Pretraining and Fine-tuning

Figure 1 presents the overall workflow of the proposed model for whole heart and brain lesion segmentation. The framework comprises three primary sections:

#### 2.2.1. PROPOSED 3D SSL STUDENT-TEACHER MODEL

A 3D student-teacher model, inspired by the 2D DINOv2 (Oquab et al., 2023) framework, is built on the xLSTM-UNet architecture for the SSL phase. We pretrained separate SSL models for cardiac and brain images, fine-tuning them for segmentation tasks. The xLSTM component captures long-range slice dependencies, ensuring spatial coherence, crucial for accurate segmentation. Unlike a Vision Transformer (ViT), our model uses xLSTM to model slice-to-slice relationships in 3D medical images, improving segmentation for cardiac and brain tasks. The student encoder is optimized via backpropagation, while the teacher encoder updates using a momentum-based EMA (Exponential Moving Average) mechanism. Contrastive self-distillation helps the student match the teacher's representations, and a hybrid loss function KL divergence and Mean Squared Error (MSE) enhances feature learning. Detailed methodology is provided in AppendixC.

#### 2.2.2. THE xLSTM MODULE

The xLSTM block integrates convolutional processing with a modified LSTM (mLSTM) for enhanced feature extraction and sequential modeling. It starts with a convolutional

layer, instance normalization (IN), and Leaky ReLU activation to capture spatial patterns and stabilize training. The output is flattened, normalized, and split into two pathways: one undergoes a linear transformation with SiLU activation, while the other undergoes a flip operation before mLSTM processing to capture long-range dependencies. The pathways are merged, followed by a final linear transformation and a residual connection to preserve information and improve gradient flow. By combining convolutional and recurrent architectures, xLSTM extracts local spatial features while efficiently modeling sequential dependencies. The flip mechanism enables bidirectional processing, ensuring both past and future dependencies are captured, while normalization and residual connections enhance stability and training efficiency.

### 2.2.3. Supervised Fine-Tuning for Segmentation

In this stage, the pre-trained SSL models on the cardiac and neurological images were fine-tuned using a limited amount of labeled data for the respective segmentation tasks, including whole heart and stroke and traumatic brain injury lesion segmentation. During this phase, the pre-trained student encoder is fine-tuned to optimize segmentation performance for specific applications.

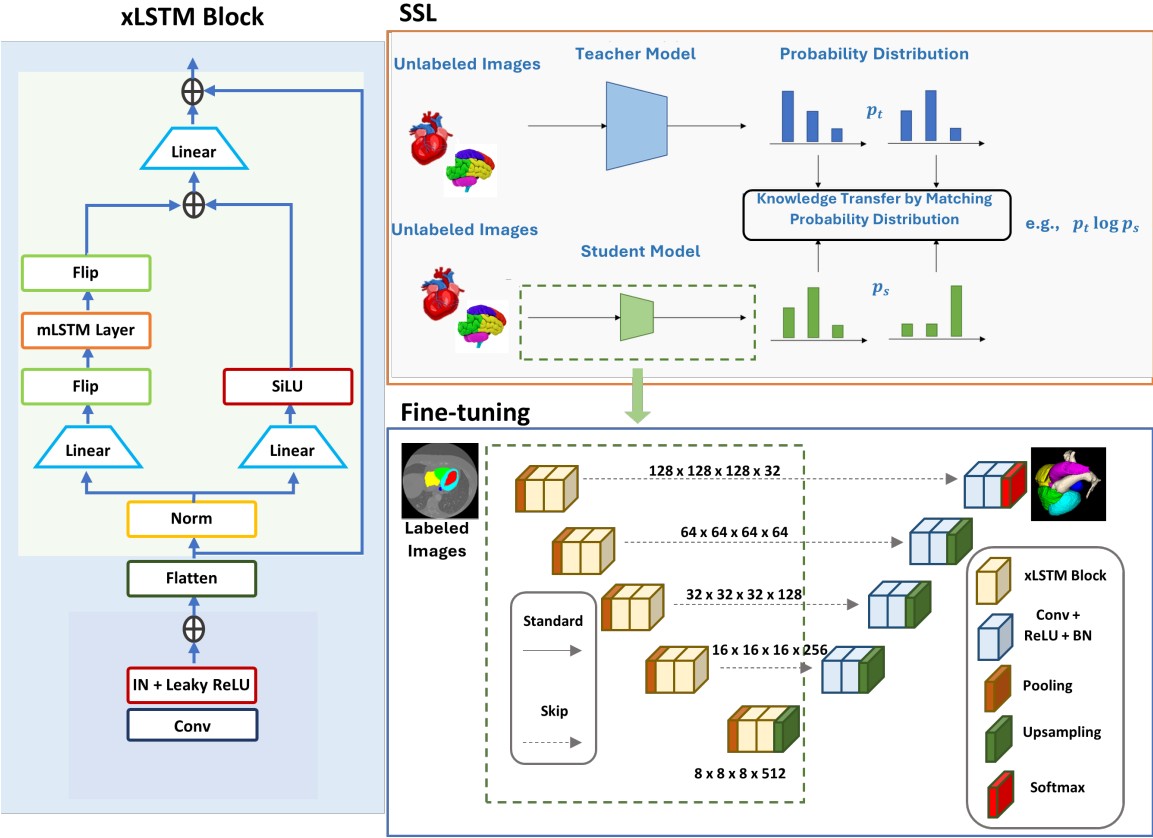

Figure 1: Overview of the 3D SSL student-teacher pretraining framework and downstream fine-tuning segmentation pipeline, incorporating the xLSTM module.

### 2.3. Evaluation and Performance Analysis

For the rigorous evaluation of the 3D-SegSync performance through comprehensive analysis, the results were benchmarked against its variant (3D-SegSync_Bottom: only bottom layer of the pretrained SSL encoder was updated) and other state-of-the-art (SOTA) models (3D-nnUNet 3D-nnUNet (Isensee et al., 2021), 3D-UNet (Ronneberger et al., 2015b), 3D-ResUNet (Li et al., 2023)), demonstrating the superior accuracy and robustness of the proposed approach in both heart and brain segmentation tasks. We further extended our comparison to the latest SSL methods that have been specifically developed for 3D medical imaging (SwinMM, SwinSSL, Voco, and Hi-End-MAE) to provide a more comprehensive evaluation of the benefits of our pretraining approach (see Table 2)

### 2.4. Training and optimization

We developed a self-supervised learning (SSL) framework in PyTorch for segmentation tasks, optimized using the Adam optimizer (LR: 0.00001) for stable convergence. Our model is trained with a batch size of 2 and a patch size of $96 \times 96 \times 96$ during SSL and $128 \times 128 \times 128$ during downstream tasks, for 1000 epochs. SSL includes data augmentation techniques such as random cropping, flipping, color jittering, Gaussian blur, and solarization, with a loss function using KL divergence and MSE. For downstream tasks, augmentations include flipping, scaling, noise addition, brightness/contrast adjustments, and RandGaussianRotate, RandGaussianSmooth, RandZoomd, RandAdjustContrast, RandGaussianNoise, RandShiftIntensity, and RandCrop. The downstream loss function combines cross-entropy and Dice loss to improve segmentation. A sliding window approach ensures smooth predictions during inference. Training took 15 hours for SSL (with early stopping at 20 epochs) and 24 hours for downstream tasks, using an A6000 GPU with 48 GB of memory.

## 3. Results

We evaluated the performance of our proposed 3D SegSync model on multiple datasets, including three whole-heart (MMWHS, WHS++, HVSMR-2.0 (Pace et al., 2024)), and two neurological/brain imaging (ISLES-2024 stroke and TBI). Results from Figure 2 consistently demonstrate that 3D-SegSync outperforms state-of-the-art (SOTA) models, achieving superior Dice scores and lower Hausdorff Distance 95% (HD95) values in all cardiac imaging datasets including CT and MRI modalities. It highlights that 3D-SegSync utilized multi-layer SSL pre-training and achieved significantly higher Dice scores and lower HD95 values compared to 3D-SegSync_bottom, which only uses SSL features from the bottom layer. This multi-layer feature extraction allows 3D-SegSync to capture richer, hierarchical representations, leading to superior segmentation performance.

In Figure 3, we present detailed segmentation results on the HVSMR-2.0 dataset to showcase 3D-SegSync's performance across all labels. The model excels in segmenting anatomical structures like the left ventricle (LV), aorta (AO), and pulmonary artery (PA), achieving superior Dice scores, even for smaller structures. This highlights the model's ability to balance local detail with broader anatomical context. Further analysis of 3D-SegSync's generalization across imaging modalities and significance maps is provided in AppendixB. This improvement stems from advanced multi-layer SSL pre-training, enabling

3D-SegSync to learn richer feature representations. Unlike 3D-SegSync-bottom, which relies on low-level features, the full 3D-SegSync model integrates high-level context for enhanced segmentation accuracy and captures fine anatomical boundaries, as indicated by significantly lower HD95 values. Figure 4 compares the performance of 3D-SegSync with state-of-the-art models (3D-SegSync-bottom, xLSTM-UNet, 3D-nnUNet, 3D-ResUNet, 3D-UNet) on the ISLES2024 stroke and TBI datasets. 3D-SegSync outperforms all models, achieving the highest Dice scores and lowest HD95 values.

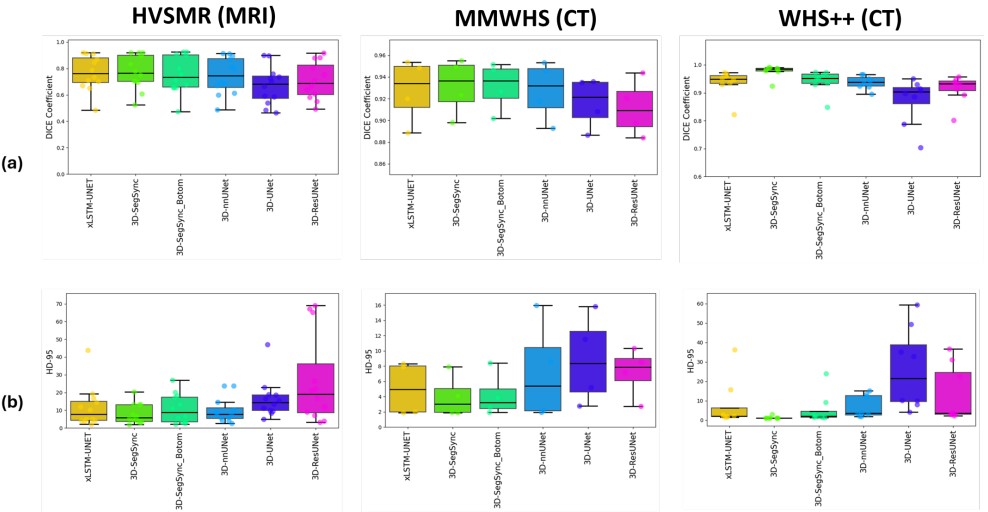

Figure 2: Performance comparison of the proposed 3D-SegSync and SOTA models on Dice and HD-95 metrics across Whole Heart segmentation datasets.

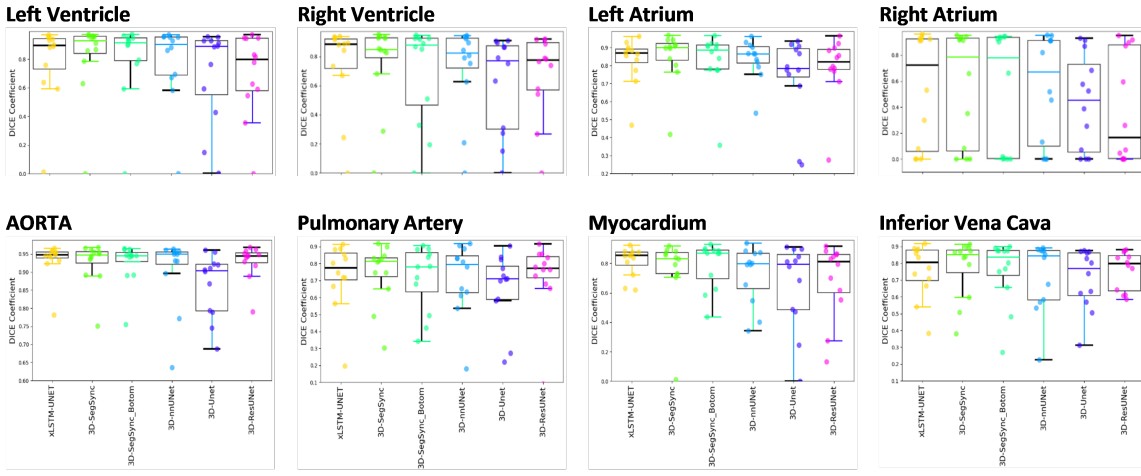

Figure 3: Dice coefficient per label for each model for performance analysis of the proposed model with SOTA approaches on the HVSMR dataset. The labels include LV, RV, and other anatomical structures.

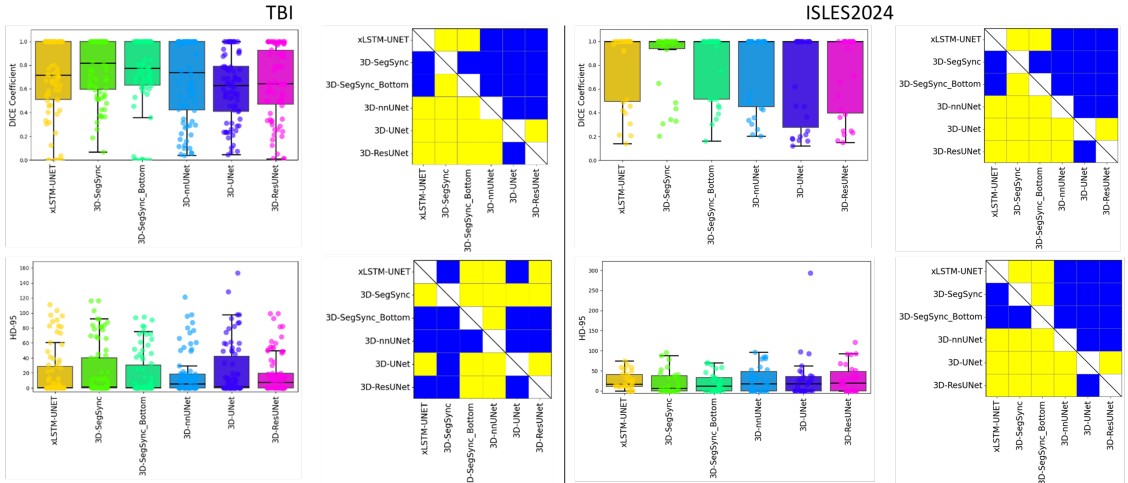

Figure 4: Performance comparison of the proposed 3D-SegSync and SOTA models on Dice and HD-95 metrics across TBI and ISLES brain lesion segmentation datasets.

Table 1: Performance analysis of 3D-SegSync with its variants and SOTA models for all heart and brain imaging datasets.

| Average Dice Coefficient (±SD) | | | | | | |
|---|---|---|---|---|---|---|
| Dataset | 3D-SegSync | 3D-SegSync-Bottom | xLSTM-UNet | 3D-muUNet | 3D-ResUNet | 3D-UNet |
| HVSMR-2.0 | **0.77 ± 0.02** | 0.75 ± 0.03 | 0.76 ± 0.04 | 0.74 ± 0.05 | 0.70 ± 0.05 | 0.67 ± 0.06 |
| MMWHS CT | **0.94 ± 0.01** | 0.93 ± 0.02 | 0.90 ± 0.03 | 0.91 ± 0.03 | 0.88 ± 0.04 | 0.85 ± 0.05 |
| MMWHS MRI | **0.88 ± 0.02** | 0.86 ± 0.03 | 0.85 ± 0.03 | 0.84 ± 0.03 | 0.81 ± 0.04 | 0.79 ± 0.05 |
| WHIS++ CT | **0.97 ± 0.01** | 0.94 ± 0.02 | 0.93 ± 0.02 | 0.92 ± 0.02 | 0.91 ± 0.03 | 0.87 ± 0.04 |
| WHIS++ MRI | **0.88 ± 0.02** | 0.87 ± 0.03 | 0.85 ± 0.03 | 0.83 ± 0.04 | 0.80 ± 0.04 | 0.78 ± 0.05 |
| TBI | **0.78 ± 0.03** | 0.72 ± 0.04 | 0.70 ± 0.04 | 0.68 ± 0.05 | 0.66 ± 0.06 | 0.63 ± 0.06 |
| ISLES2024 | **0.84 ± 0.02** | 0.80 ± 0.03 | 0.79 ± 0.03 | 0.76 ± 0.04 | 0.74 ± 0.04 | 0.72 ± 0.05 |
| Average HD (±SD) | | | | | | |
| Dataset | 3D-SegSync | 3D-SegSync-Bottom | xLSTM-UNet | 3D-muUNet | 3D-ResUNet | 3D-UNet |
| HVSMR-2.0 | **17.25 ± 2.4** | 24.21 ± 3.2 | 21.19 ± 2.8 | 22.16 ± 3.1 | 28.87 ± 3.7 | 33.17 ± 4.0 |
| MMWHS CT | **14.39 ± 1.2** | 18.62 ± 2.5 | 19.56 ± 2.6 | 19.14 ± 2.8 | 35.48 ± 4.1 | 38.25 ± 4.5 |
| MMWHS MRI | **29.02 ± 3.1** | 31.76 ± 3.6 | 33.41 ± 4.0 | 34.12 ± 4.1 | 38.62 ± 4.7 | 40.77 ± 5.1 |
| WHIS++ CT | **5.28 ± 0.8** | 17.94 ± 2.2 | 21.58 ± 2.4 | 29.60 ± 3.5 | 61.24 ± 5.2 | 65.11 ± 5.6 |
| WHIS++ MRI | **5.12 ± 0.9** | 13.17 ± 1.7 | 21.43 ± 2.3 | 25.01 ± 2.9 | 58.71 ± 5.0 | 62.88 ± 5.4 |
| TBI | **19.45 ± 2.6** | 23.17 ± 3.1 | 24.13 ± 3.4 | 27.20 ± 3.9 | 33.57 ± 4.3 | 36.12 ± 4.8 |
| ISLES2024 | **29.22 ± 3.2** | 31.67 ± 3.5 | 34.72 ± 3.8 | 35.09 ± 4.0 | 39.61 ± 4.5 | 38.87 ± 4.7 |

Table 1 shows that 3D-SegSync outperforms all models in heart and brain imaging datasets, with 3D-SegSync-bottom coming second. This variant fine-tunes only the bottom-layer features, indicating that optimizing all encoder layers improves performance. For a comprehensive comparison with the latest 3D SSL SOTA models (Hi-End-MAE, SwinSSL, SwinMM, and Voco), Table 2 presents performance scores on the MMWHS (CT) dataset for whole heart segmentation. 3D-SegSync achieves the highest Dice score (0.94), lowest HD (14.39), HD95 (4.197), ASSD (0.942), and Vol Diff (0.0062), outperforming other models. Compared to SwinSSL (Dice: 0.909, HD: 19.033) and Hi-End-MAE (Dice: 0.869, HD: 23.212), our model demonstrates superior segmentation accuracy and robustness, highlighting its effectiveness in cardiac medical image analysis. A further explanation of the results of each dataset can be found in AppendixB.

Table 2: Performance analysis of 3D-SegSync with latest SOTA 3D SSL models for MMWHS(CT) dataset.

| Model | Dice (±SD) | HD (±SD) | HD95 (±SD) | ASSD (±SD) | Vol_Diff (±SD) | p-value (Dice) |
|---|---|---|---|---|---|---|
| 3D-SegSync | **0.94 ± 0.01** | **14.39 ± 1.2** | **4.197 ± 0.4** | **0.942 ± 0.1** | **0.0062 ± 0.0005** | **-** |
| SwinMM (2023) | 0.91 ± 0.03 | 17.33 ± 2.8 | 6.762 ± 1.4 | 1.087 ± 0.3 | 0.00785 ± 0.0010 | 0.002* |
| SwinSSL (2022) | 0.90 ± 0.04 | 19.03 ± 3.1 | 8.011 ± 1.7 | 2.089 ± 0.5 | 0.00922 ± 0.0012 | 0.001* |
| Voco (2024) | 0.92 ± 0.02 | 16.78 ± 2.5 | 6.181 ± 1.3 | 1.111 ± 0.3 | 0.00779 ± 0.0009 | 0.003* |
| Hi-End-MAE (2025) | 0.86 ± 0.05 | 23.21 ± 3.5 | 10.81 ± 2.0 | 2.009 ± 0.6 | 0.00982 ± 0.0015 | <0.001* |

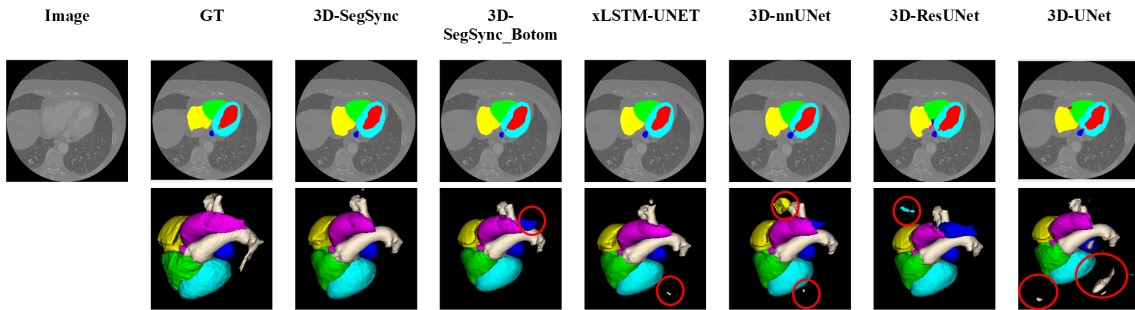

Figure 5: Quantitative Performance of the Proposed and SOTA Models on the MMWHS CT Dataset. Colour representation: Purple (AO), Yellow (RA), Red (LV), Light Blue (Myo), Gray (PA), Blue (LA), Green (RV).

Figure 5 illustrates the quantitative performance of the proposed 3D-SegSync on the MMWHS_CT dataset. The results demonstrate that the proposed 3D-SegSync model achieves a segmentation output that is closely aligned with the ground truth (GT) segmentation map, outperforming other SOTA models across most anatomical regions of the whole heart. Among the comparative models, 3D-ResUNet and 3D-UNet demonstrate a higher rate of segmentation errors, especially in the pulmonary veins and aorta. The SOTA 3D-nnUNet, while performing comparatively better, exhibits noticeable errors in the right atrium, as shown in the 3D segmentation map. These observations provide valuable insights into the potential areas for further refinement in cardiac segmentation methods.

Beyond whole-heart segmentation, we validated the efficacy of our proposed model on the neurological/brain imaging datasets such as the TBI dataset (see Figure 4) from the MICCAI (Medical Image Computing and Computer Assisted Intervention) 2024 Grand Challenge. Our model secured first place in the TBI validation and testing phases, demonstrating its exceptional accuracy and generalisability. The leaderboard for the TBI challenge can be accessed at https://aims-tbi.grand-challenge, where our team, DeepLearnAI, is listed in the top position. Additionally, we tested our model on the ISLES-2024 stroke challenge (see Figure 4), achieving first place on the leaderboard under the team name Dolphins. The leaderboard for the ISLES challenge can be viewed at https://isles-24.grand. These achievements on both TBI and ISLES-2024 challenges underline the superior performance of our proposed model compared to other SOTA deep learning approaches.

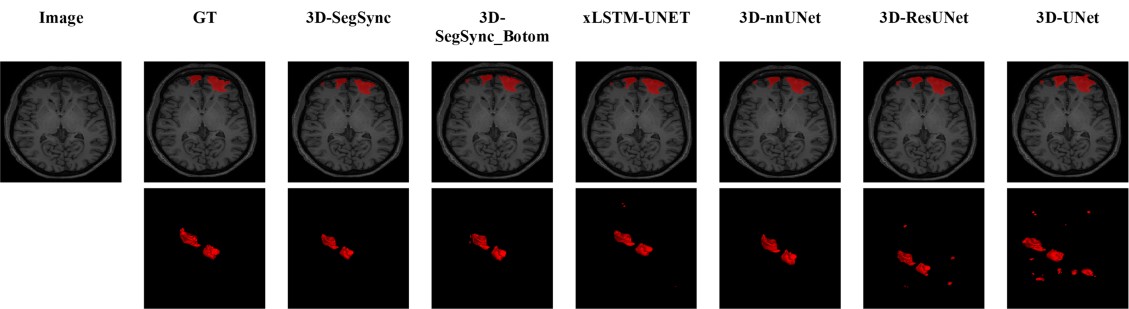

Figure 6: Quantitative analysis of the proposed 3D-SegSync model compared to state-of-the-art (SOTA) models for TBI lesion segmentation.

Figure 6 shows a quantitative analysis of the 3D-SegSync model for TBI lesion segmentation, demonstrating its superior accuracy compared to state-of-the-art models. Unlike the baseline xLSTM model, 3D-SegSync captures the intricate features of moderate to severe TBI (msTBI) lesions, overcoming challenges of high variability for precise segmentation. Its success across various datasets and modalities highlights its robustness and versatility. By leveraging pre-trained SSL features, the model reduces dependence on large labeled datasets, making it highly effective in medical imaging with limited annotated data. These results position 3D-SegSync as a reliable solution for medical image segmentation.

Future work could involve applying SSL to larger datasets for improved generalizability, extending 3D-SegSync to other imaging modalities (e.g., ultrasound, PET), and incorporating multi-modal data (e.g., clinical or genomic data) to improve diagnostic accuracy. Incorporating interpretability techniques could further enhance trust in clinical applications. Addressing these areas will help 3D-SegSync evolve into a more powerful tool for medical imaging.

## 4. Conclusion

We introduced 3D-SegSync, a robust 3D medical image segmentation framework designed to address challenges like data scarcity, modality variability, and anatomical complexity. By combining the DINOv2 teacher-student architecture with the xLSTM-UNet, 3D-SegSync leverages self-supervised learning to extract rich, modality-independent 3D features from large-scale unlabeled datasets. The xLSTM-UNet further enhances the model's ability to capture spatial and contextual relationships in 3D imaging, making it highly effective for segmentation tasks. This fully 3D framework achieves state-of-the-art performance in whole-heart, stroke lesion, and traumatic brain injury segmentation across CT and MRI, significantly reducing dependence on labeled datasets. By uniting powerful 3D self-supervised learning with efficient design, 3D-SegSync sets a new benchmark in medical imaging, offering improved scalability and clinical relevance. Future work will explore its application to broader tasks, strengthening its cross-modality capabilities and impact.

## Acknowledgments

We would like to express our gratitude to all those who contributed to this work. Their insightful feedback, support, and collaboration were invaluable.

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

## Appendix A. Proposed Framework

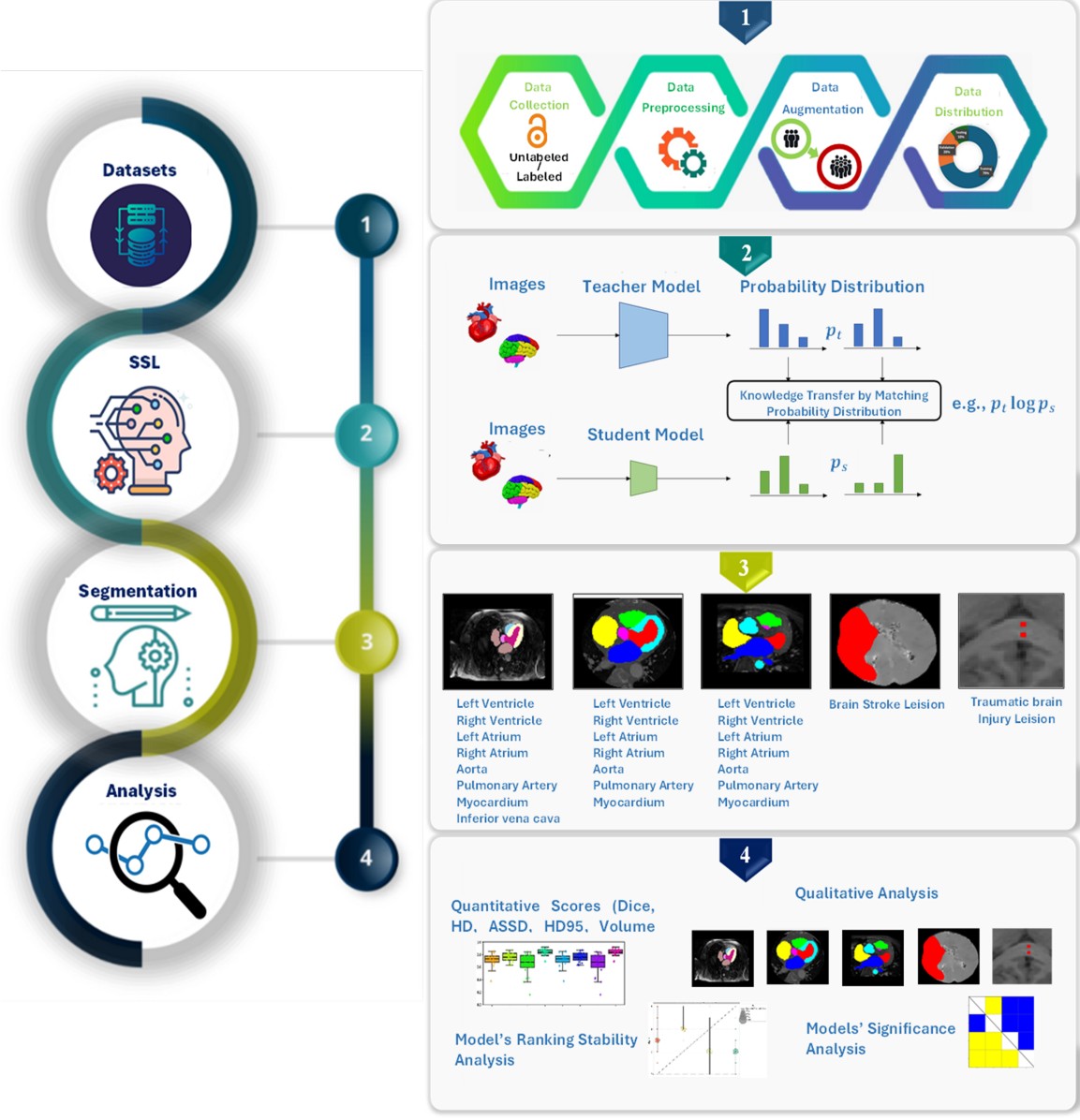

Figure 7: 3D-SegSync framework architecture ( 1. data curation, 2. pretraining using SSL on unlabelled datasets on cardiac and brain images, 3. fine-tuning step for cardiac and brain image segmentation using labeled datasets, 4. Performance analysis of 3D-Segsync with SOTA models).

## Appendix B. Comparsions of 3D-SegSync with SOTA models

Figure 9 demonstrates 3D-SegSync's ability to generalize across different imaging modalities, outperforming SOTA models in both CT and MRI datasets for whole-heart segmentation. This ability to learn modality-independent features via SSL pre-training ensures its applicability in clinical settings where multimodal imaging is common.

Figure 8 presents significance maps, where 3D-SegSync consistently shows higher yellow regions, indicating statistically significant improvements in Dice and HD95 metrics compared to other models. 3D SegSync_bottom shows fewer yellow regions in comparison with its advance version 3D-SegSync, reflecting its weaker performance, while 3D-UNet displays highest blue regions, indicating significantly lowest performance among all models in most of the datasets such as HVSMR and WHS++. We have also given the model ranks on the all whole heart segmentation datasets in Figure 10.

Table 3: Performance analysis of proposed and state of the art models for HVSMR-2.0 dataset.

| Model | Dice_Avg_All | HD_Avg_All | HD95_Avg_All | ASSD_Avg_All | Vol_Diff_Avg_All |
|---|---|---|---|---|---|
| 3D-SegSync | 0.77132 | 17.25634 | 8.006959 | 2.02056 | 0.02273048 |
| 3D-SegSync_Botom | 0.753696 | 24.21755 | 10.89928 | 3.03842 | 0.024452868 |
| xLSTM-UNET | 0.766224 | 29.22956 | 11.74866 | 3.100872 | 0.026769222 |
| 3D-nnUNet | 0.745084 | 22.12653 | 10.00376 | 2.39672 | 0.026893855 |
| 3D-ResUNet | 0.706659 | 68.07598 | 26.91917 | 7.344615 | 0.037925875 |
| 3D-UNet | 0.670413 | 35.08616 | 16.39909 | 4.138591 | 0.037455937 |

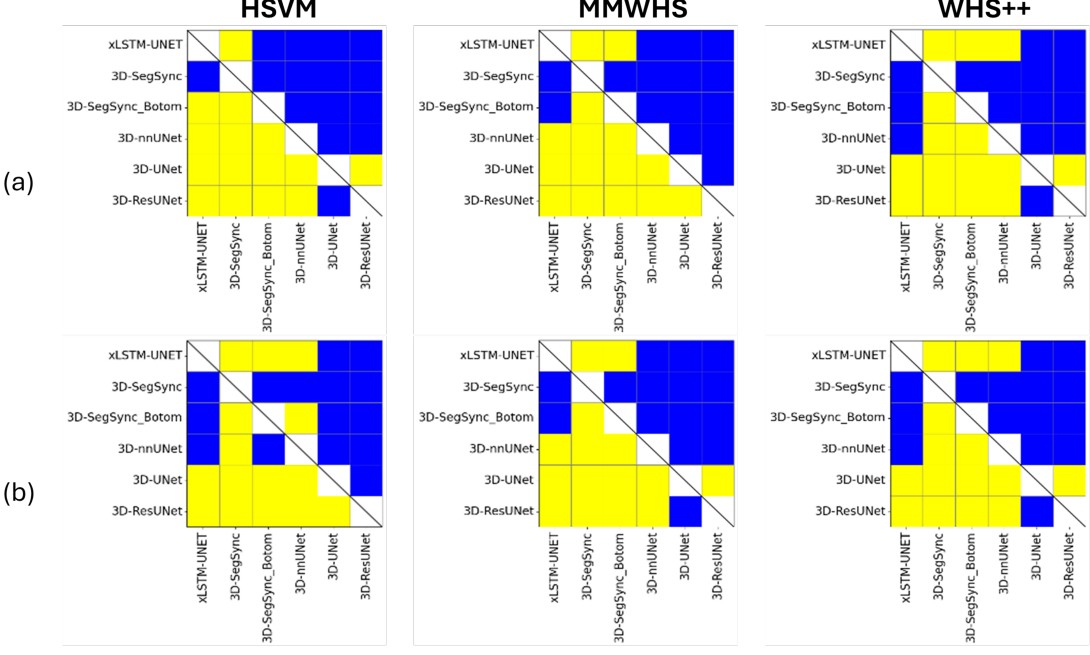

Figure 8: Significance maps of the proposed 3D-SegSync and SOTA models on DICE (a) and HD-95 (b) metrics across Whole Heart segmentation datasets.

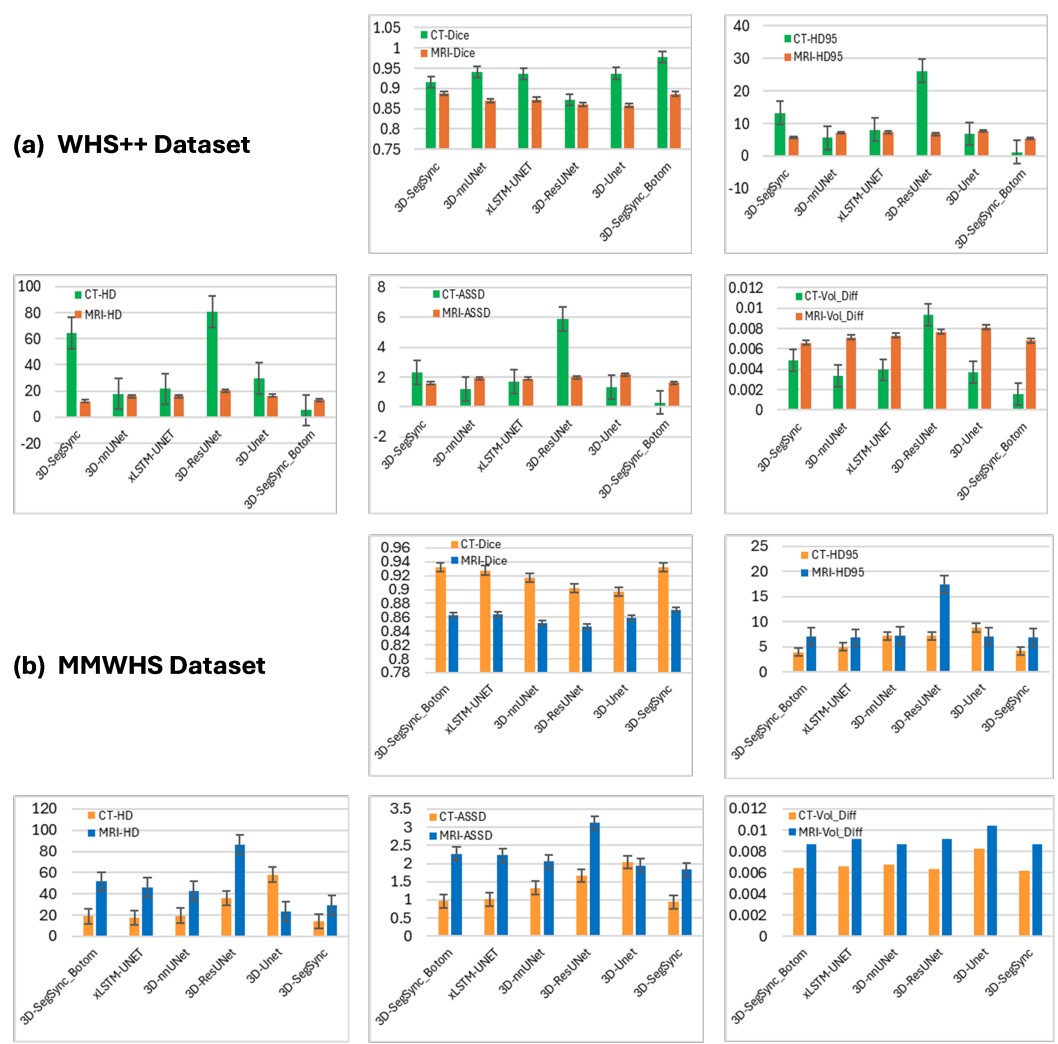

Figure 9: Cross modality performance comparison of 3D-Segsync and SOTA models for whole-heart how segmentation across CT and MRI datasets. (a) WHS++ dataset, where green bars show CT and mustard bars show MRI. (b) MMWHS dataset, where orange bars show CT and blue bars show MRI.

Table 3 shows the segmentation results on the HVSMR-2.0 whole-heart MRI dataset demonstrate that 3D-SegSync achieved the best overall performance, with the highest Dice score (0.7713), the lowest ASSD (2.0206 mm), and the smallest volume difference (0.0227), indicating accurate overlap, surface alignment, and volume estimation. 3D-SegSync_Botom and xLSTM-UNET also performed well, with Dice scores of 0.7537 and 0.7662, respectively, though both exhibited higher Hausdorff distances (HD95 of 10.8993 mm and 11.7487 mm) and ASSD values, reflecting less precise boundary alignment. While 3D-nnUNet showed good surface alignment (ASSD of 2.3967 mm), its lower Dice score (0.7451) and higher HD95 (10.0038 mm) suggest moderate segmentation accuracy. In contrast, 3D-ResUNet

and 3D-UNet underperformed, with significantly lower Dice scores (0.7067 and 0.6704) and much higher HD95 (26.9192 mm and 16.3991 mm), indicating poor boundary and surface alignment. Overall, 3D-SegSync is the most reliable model for whole-heart segmentation in this dataset.

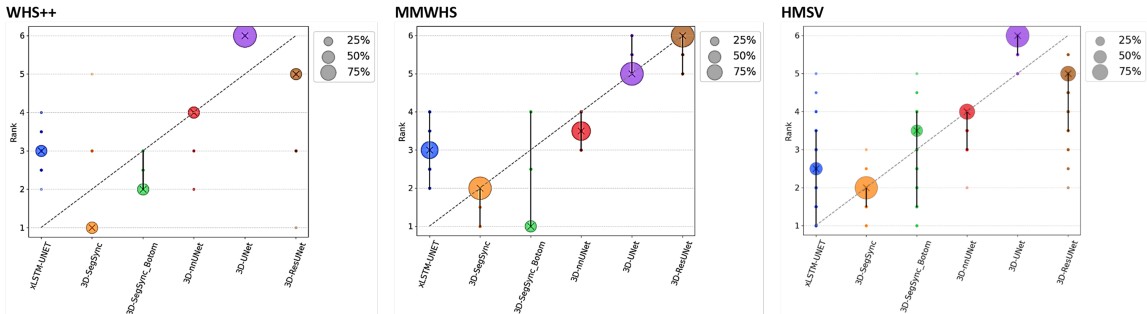

Figure 10: Blob plots illustrating the stability of rankings of whole heart segmentation datasets based on bootstrap sampling. The median rank for each algorithm is represented by a black cross, while the 95% bootstrap intervals across samples are depicted as black lines.

Table 4 shows the segmentation results on the MMWHS dataset demonstrate that 3D-SegSync achieved the best overall performance, with the highest Dice score (0.9415), the lowest ASSD (0.9425 mm), and a minimal volume difference (0.0062), indicating excellent overlap, surface alignment, and volume estimation. 3D-SegSync-Botom also performed well, with a Dice score of 0.9316 and the lowest HD95 (3.9579 mm), though it showed slightly higher ASSD (0.9644 mm). xLSTM-UNET attained a Dice score of 0.9277 but exhibited higher HD95 (5.0222 mm) and ASSD (1.0150 mm), reflecting less precise boundary alignment. 3D-nnUNet demonstrated moderate performance, with a Dice score of 0.9175 and higher HD95 (7.1660 mm) and ASSD (1.3304 mm). In contrast, 3D-ResUNet and 3D-UNet underperformed, with significantly lower Dice scores (0.8916 and 0.8864) and considerably higher HD95 (7.2059 mm and 8.8210 mm) and ASSD (1.6663 mm and 2.0362 mm), indicating poor boundary and surface alignment. Overall, 3D-SegSync is the most effective model for whole-heart segmentation in this dataset.

Table 4: Performance analysis of proposed and SOTA models using MMWHS CT dataset.

| Model | Dice_Avg | HD_Avg | HD95_Avg | ASSD_Avg | Vol_Diff_Avg |
|---|---|---|---|---|---|
| 3D-SegSync | 0.941531281 | 14.3973592 | 4.197811849 | 0.942519917 | 0.006214206 |
| 3D-SegSync-Botom | 0.93157309 | 18.8680027 | 3.95788033 | 0.964365257 | 0.006409229 |
| xLSTM-UNET | 0.927654749 | 17.6516858 | 5.022211073 | 1.014969103 | 0.006599423 |
| 3D-nnUNet | 0.917473901 | 19.6456178 | 7.166018194 | 1.330353153 | 0.006746526 |
| 3D-ResUNet | 0.891598521 | 35.883997 | 7.205884387 | 1.666301144 | 0.006330065 |
| 3D-UNet | 0.886358506 | 58.1659145 | 8.820965598 | 2.036248398 | 0.008241135 |

Tables 5, 6, and 7 present the results of CT and MRI whole-heart segmentation, where the proposed 3D-SegSync model, employing a self-supervised learning approach, consistently outperformed state-of-the-art (SOTA) models across Dice and other key performance

metrics. The model demonstrated superior accuracy in overlap, boundary alignment, and volume estimation, validating its effectiveness for both CT and MRI modalities. Additionally, the model was evaluated on the WHS++ dataset, where similar performance trends were observed, reinforcing its generalisability to different datasets.

Furthermore, the proposed 3D-SegSync was assessed against SOTA models for stroke lesion segmentation (ISLES2024) and traumatic brain injury (TBI) lesion segmentation tasks. In these evaluations (Tables 8 and 9), 3D-SegSync consistently delivered robust and reliable performance, showcasing its versatility and efficacy in segmenting diverse anatomical and pathological structures. These results highlight the potential of the self-supervised 3D-SegSync model to set a new standard in medical image segmentation across multiple domains.

Table 5: Performance analysis of proposed and SOTA models using MMWHS MRI dataset.

| Model | Dice_Avg | HD_Avg | HD95_Avg | ASSD_Avg | Vol_Diff_Avg |
|---|---|---|---|---|---|
| 3D-SegSync | 0.87167 | 29.02241 | 6.871293 | 1.831784 | 0.008656418 |
| 3D-SegSync_Botom | 0.86413 | 46.32272 | 6.813768 | 2.231868 | 0.009208905 |
| xLSTM-UNET | 0.86338 | 51.50929 | 7.029732 | 2.266159 | 0.008637976 |
| 3D-nnUNet | 0.85904 | 23.19734 | 7.03841 | 1.946138 | 0.010388217 |
| 3D-ResUNet | 0.84187 | 42.48808 | 7.26451 | 2.05212 | 0.00868741 |
| 3D-UNet | 0.83663 | 86.60793 | 17.44347 | 3.115857 | 0.009206529 |

Table 6: Performance analysis of proposed and SOTA models using WHS++ CT dataset.

| Model | Dice_Avg | HD_Avg | HD95_Avg | ASSD_Avg | Vol_Diff_Avg |
|---|---|---|---|---|---|
| 3D-SegSync | 0.976996 | 5.281255 | 1.251489 | 0.301377 | 0.001588615 |
| 3D-SegSync_Botom | 0.941146 | 17.94436 | 5.572338 | 1.184542 | 0.003344029 |
| xLSTM-UNET | 0.935651 | 21.58496 | 8.052558 | 1.677745 | 0.003928403 |
| 3D-nnUNet | 0.927362 | 29.60166 | 6.834203 | 1.301429 | 0.003741077 |
| 3D-ResUNet | 0.915044 | 64.24364 | 13.20491 | 2.291552 | 0.004881045 |
| 3D-UNet | 0.872164 | 80.54156 | 26.13814 | 5.87906 | 0.00934523 |

Table 7: Performance analysis of proposed and SOTA models using WHS++ MRI dataset.

| Model | Dice_Avg | HD_Avg | HD95_Avg | ASSD_Avg | Vol_Diff_Avg |
|---|---|---|---|---|---|
| 3D-SegSync | 0.887617 | 12.19138 | 5.696797 | 1.577587 | 0.006581231 |
| 3D-SegSync_Botom | 0.886642 | 13.17895 | 5.460724 | 1.625685 | 0.006799575 |
| xLSTM-UNET | 0.87293 | 16.08778 | 7.373174 | 1.892255 | 0.007296956 |
| 3D-nnUNet | 0.869232 | 16.02272 | 7.136496 | 1.927073 | 0.007132139 |
| 3D-ResUNet | 0.851003 | 20.14479 | 6.741368 | 1.969676 | 0.007659463 |
| 3D-UNet | 0.858753 | 16.35387 | 7.723141 | 2.168973 | 0.008124995 |

Table 8: Performance analysis of proposed and SOTA models using TBI MRI dataset.

| Model | Dice_Avg | HD_Avg | HD95_Avg | ASSD_Avg | Vol_Diff_Avg |
|---|---|---|---|---|---|
| 3D-SegSync | 0.782532 | 19.45671 | 15.36666 | 2.45671 | 0.275341 |
| 3D-SegSync_Botom | 0.724659 | 23.17893 | 17.99199 | 3.01345 | 0.492328 |
| xLSTM-UNET | 0.686121 | 24.13334 | 19.92581 | 3.18965 | 0.718707 |
| 3D-nnUNet | 0.678905 | 25.31134 | 17.84589 | 3.30567 | 0.739339 |
| 3D-ResUNet | 0.678905 | 27.21234 | 17.84589 | 4.17865 | 0.786737 |
| 3D-UNet | 0.643323 | 28.18971 | 18.62511 | 6.34567 | 0.852835 |

Table 9: Performance analysis of proposed and SOTA models using ISLES2024 dataset.

| Model | Dice_Avg | HD_Avg | HD95_Avg | ASSD_Avg | Vol_Diff_Avg |
|---|---|---|---|---|---|
| 3D-SegSync | 0.848296 | 29.22114 | 21.05371 | 2.78653 | 0.840097 |
| 3D-SegSync_Botom | 0.804858 | 31.67891 | 20.2839 | 1.89765 | 0.840068 |
| xLSTM-UNET | 0.798529 | 34.72802 | 25.56568 | 2.93112 | 0.865685 |
| 3D-nnUNet | 0.769919 | 35.09987 | 27.30891 | 2.98765 | 0.887896 |
| 3D-ResUNet | 0.74243459 | 39.61133 | 30.11339 | 4.23478 | 0.900569 |
| 3D-UNet | 0.702374 | 38.87633 | 30.9896 | 6.13459 | 0.946751 |

## Appendix C. Methodology & Mathematical Details of the Proposed Framework

### C.1. Methodology

The proposed framework is built on a self-supervised learning (SSL) (Mazher et al., 2024) approach designed to pre-train a 3D Vision-LSTM (xLSTM) integrated UNet model (xLSTM-UNet) (Oquab et al., 2023; Chen et al., 2024). The methodology combines advanced deep learning techniques to achieve enhanced performance in 3D medical image segmentation tasks. The main diagram of proposed SSL model is shown in AppendixA.

#### C.1.1. Data Augmentation in the Student-Teacher Framework

Robust data augmentation plays a critical role in the SSL pipeline. Techniques such as flipping, scaling, Gaussian noise addition, Gaussian blur, and adjustments to brightness and contrast are applied to create diverse and informative training inputs. Two augmented views of each input image are generated and processed through a Siamese network structure, comprising the student and teacher encoders. The teacher encoder's outputs are refined through centring, sharpening, and normalisation via a softmax function, producing supervision signals for the student encoder.

The loss function ensures alignment between the student's outputs and the teacher's processed outputs by minimising divergence, employing cross-entropy loss and mean squared error (MSE) (Oquab et al., 2023). This alignment facilitates robust feature learning from unlabelled data, enhancing the model's generalisation capabilities.

### C.1.2. xLSTM-UNet Architecture

The xLSTM-UNet model (Chen et al., 2024) integrates Vision-LSTM (xLSTM), an advanced extension of Long Short-Term Memory (LSTM) networks, into the UNet architecture. xLSTM excels at capturing long-range dependencies and contextual information, complementing the UNet's strength in extracting local features through its convolutional encoder-decoder design. The encoder identifies hierarchical features from the input, while the decoder reconstructs these features into detailed segmentation maps, enabling precise and reliable segmentation.

### C.1.3. Self-Supervised Pre-Training and Supervised Fine-Tuning

The SSL framework focuses on pre-training the xLSTM-UNet encoder using unlabelled data to capture meaningful spatial and contextual features. Once pre-trained, the encoder is fine-tuned in a supervised manner using labelled datasets, optimising the decoder to generate accurate segmentation maps. This two-stage process minimises the reliance on extensive labelled datasets, while the xLSTM module ensures effective learning of global context and long-range dependencies.

## C.2. Mathematical Details of the Proposed Framework

The momentum teacher encoder's parameters $\theta_t$ are updated based on the student encoder's parameters $\theta_s$ using a momentum-based approach:

$$\theta_t = m \cdot \theta_t + (1 - m) \cdot \theta_s \tag{1}$$

Where $\theta_t$ are the parameters of the teach encoder, $\theta_s$ are the parameters of the student encoder, $m$ is the momentum coefficient typically a value close to 1.

Let $x$ be the original input image. Two different views of the input, $x_1$ and, $x_2$ are generated using strong data augmentations:

$$x_1 = \text{Augment}(x), \quad x_2 = \text{Augment}(x) \tag{2}$$

Both views are then processed through the student encoder $f_s$ and teacher encoder $f_t$ to extract feature representations:

$$h_1 = f_s(x_1; \theta_s), \quad h_2 = f_s(x_2; \theta_s) \tag{3}$$

$$h_1' = f_s(x_1; \theta_t), \quad h_2' = f_s(x_2; \theta_t) \tag{4}$$

Where $h_1$ and $h_2$ are the feature representations from the student encoder, $h_1'$ and $h_2'$ are the feature representations from the teacher encoder.

The feature representations $h_1$, $h_2$, $h_1'$, $h_2'$ are subjected to global average pooling to reduce them into feature vectors:

$$v_1 = \text{GAP}(h_1), \quad v_2 = \text{GAP}(h_2) \tag{5}$$

$$v_1' = \text{GAP}(h_1'), \quad v_2' = \text{GAP}(h_2') \tag{6}$$

Where $v_1$, $v_2$, $v_1'$ and $v_2'$ are the resulting feature vectors.

$$z_1 = \text{MLP}(v_1), \quad z_2 = \text{MLP}(v_2) \tag{7}$$

$$z_1' = \text{MLP}(v_1'), \quad z_2' = \text{MLP}(v_2') \tag{8}$$

After projection, the teacher's output is centered, sharpened, and passed through a softmax function to produce the supervision signal:

$$q_1' = \text{Softmax}\left(\frac{\text{Center}(z_1')}{\tau}\right) \tag{9}$$

$$q_2' = \text{Softmax}\left(\frac{\text{Center}(z_2')}{\tau}\right) \tag{10}$$

Where $Center(z)$ subtracts the mean of the vector to have zero mean. $\tau$ is the temperature parameter controlling the sharpness of the distribution. $Softmax(z)$ normalizes the vector into a probability distribution.

The loss function is designed to minimize the divergence between the student's feature vectors and the teacher's processed outputs. A common choice is the cross-entropy loss or mean squared error (MSE) between the student's and teacher's outputs:

$$L = \frac{1}{2}\left(\text{Loss}(z_1, q_2') + \text{Loss}(z_2, q_1')\right) \tag{11}$$

Where this loss function encourages the student encoder to produce feature representations that align closely with the teacher's outputs, thus enabling effective learning from the unlabeled data.

For the downstreaming segmentation training task we used the cross-entropy loss as given in the equation below :

$$L(z, q') = -\sum_{k=1}^{K} q'[k] \log(\text{Softmax}(z)[k]) \tag{12}$$

