# OpenReview forum: "Advancing Medical Image Segmentation with Self-Supervised Learning: A 3D Student-Teacher Approach for Cardiac and Neurological Imaging"
_MIDL.io/2025/Conference — MIDL 2025 Poster_

### Official Review · Reviewer_pTDs · 2025-02-19

**Confidence:** 4
**Preliminary Rating:** 3
**Final Rating:** 3

**Summary:**

The paper introduces 3D-SegSync, a self-supervised learning (SSL) framework that utilizes the DINOv2 pretraining paradigm with a 3D Vision-LSTM (xLSTM) backbone. 3D-SegSync is pretrained on a large-scale dataset and fine-tuned on downstream labeled data, achieving state-of-the-art performance in cardiac and brain image segmentation.

**Strengths:**

The paper is well-motivated and well-structured, with clear and easy-to-follow writing. The experiment is comprehensive, involving multiple baseline networks and effectively demonstrating the method's effectiveness through both quantitative and qualitative analyses.

**Weaknesses:**

The technical innovation is limited.

The rationale for choosing DINOv2 with the 3D Vision-LSTM (xLSTM) backbone is not clearly justified.

Additionally, the paper makes several strong claims in the introduction that are not sufficiently supported in the later sections.

**Detailed Comments:**

While the pretrained model demonstrates effectiveness on downstream tasks, its technical innovation is limited. Both DINOv2 and 3D Vision-LSTM (xLSTM) are not the primary technical contributions. The authors should emphasize their technical novelties, such as any new designs introduced when integrating DINOv2 into 3D or combining DINOv2 with the chosen backbone.

The authors should justify why the DINOv2 SSL pretraining strategy was chosen over other SSL methods. Numerous SSL approaches have been specifically developed for 3D medical image analysis recently, and including them as baselines would be beneficial.

In the abstract, the authors claim that 3D-SegSync is robust and modality-agnostic. However, there is little evidence in the later sections to substantiate these claims.

**Justification Of The Final Rating:**

Thank you for the clarifications. While the authors have justified the use of DINOv2, the lack of a baseline and statistical analysis remains a major concern. Given the current shapeof the paper, I will maintain my original rating.

**Justification Of The Preliminary Rating:**

While the paper presents clear motivation and effective results, the lack of methodological innovation remains a major flaw. The authors should justify why DINOv2 is preferable over other SSL methods, and comparing it with additional baseline SSL methods would strengthen the paper. The preliminary rating is based on the paper's current presentation.

**Questions To Address In The Rebuttal:**

Please check the Weaknesses* and Detailed Comments*.

**Special Issue:**

No

---

> ### Author Response · Authors · 2025-03-07
> **Clarifications and Justifications for Technical Innovations, SSL Strategy, and Comparisons**
>
> Q1: While the pretrained model demonstrates effectiveness on downstream tasks, its technical innovation is limited. Both DINOv2 and 3D Vision-LSTM (xLSTM) are not the primary technical contributions. The authors should emphasize their technical novelties, such as any new designs introduced when integrating DINOv2 into 3D or combining DINOv2 with the chosen backbone.
> R1:Thank you for your valuable feedback. We appreciate your concerns regarding the technical innovation of our work. To clarify, we have integrated an SSL approach inspired by the DINOv2 framework with 3D Vision-LSTM (xLSTM). We have explained the rationale behind our choice of DINOv2 pretraining and its advantages over other SSL methods for 3D medical imaging in the manuscript.
>
> The DINOv2 SSL model, originally designed for 2D natural images using a student-teacher approach with Vision Transformers (ViTs), is adapted for 3D volumetric data like CT and MRI scans in our work. The key novelty lies not in DINOv2 itself but in the extension of this approach to the 3D domain, specifically for medical image segmentation tasks. While DINOv2 originally uses Vision Transformers (ViTs) for learning self-supervised representations in 2D framework, our approach leverages the same student-teacher paradigm but with an important modification by utilizing the 3D xLSTM-based encoder instead of vision transformers. An xLSTM block-based encoder can be more effective than a Vision Transformer (ViT)-based encoder when working with 3D medical images, particularly for tasks involving sequential or spatial patterns. 3D medical images, like MRI or CT scans, consist of multiple slices, and these slices often contain relationships that need to be modeled across the volume. xLSTMs are great at capturing these sequential dependencies and spatial relationships within the slices, making them efficient for handling 3D volumes. They also require fewer resources like computation and memory, which is important when dealing with medical datasets that may not be as large as those in other domains. We have added more information in the manuscript.
>
> Q2: The authors should justify why the DINOv2 SSL pretraining strategy was chosen over other SSL methods.
> R2:We chose the DINOv2-inspired pretraining strategy for its efficiency in 3D SSL. Unlike other methods like SimCLR, MoCo, and MAE, DINO’s momentum teacher-student framework avoids the need for negative samples or complex memory banks, making it more memory- and computationally efficient. It also maintains spatial integrity, crucial for 3D medical images, without using masking strategies like MAE. DINO’s distillation mechanism ensures stable training and robust representations. Compared to other 3D SSL methods like SwinUNETR, our approach captures long-range slice-to-slice dependencies and global spatial relationships, enhanced by xLSTM, making it well-suited for medical image segmentation. We have updated the introduction to explain this.
>
> Q3: Numerous SSL approaches have been specifically developed for 3D medical image analysis recently, and including them as baselines would be beneficial.
> R3: Thank you for your thoughtful feedback. We acknowledge the importance of comparing our model with other state-of-the-art SSL methods that have been specifically developed for 3D medical imaging. Although we have already compared our method to leading segmentation models, such as SwinUNETR and other state-of-the-art methods, we agree that including comparisons with recent SSL methods designed for 3D medical imaging (like SwinMM, SwinSSL, Voco, and Hi-End-MAE) would provide a more comprehensive evaluation of the benefits of our pretraining approach. We have added a comparison of these 3D SSL SOTA models in the manuscript with our proposed 3D-SegSync (please see Table 2)
>
> Q4: While the paper presents clear motivation and effective results, the lack of methodological innovation remains a major flaw. The authors should justify why DINOv2 is preferable over other SSL methods and comparing it with additional baseline SSL methods would strengthen the paper. The preliminary rating is based on the paper's current presentation.
> R4: Thank you for your valuable feedback and comments. We fully agree with your points and have updated the manuscript to clarify the motivation behind adopting the DINOv2 student-teacher structure. Additionally, we have included more results comparing our proposed 3D-SegSync with the latest 3D SSL methods to demonstrate its performance.
>
> Q5: In the abstract, the authors claim that 3D-SegSync is robust and modality-agnostic. However, there is little evidence in the later sections to substantiate these claims.
> R5: We appreciate your insightful feedback and suggestions. We try to limit our claim in the abstract and make it more concise.

---

> > ### Comment · Reviewer_pTDs · 2025-03-13
> >
> > I would like to extend my appreciation to the authors for the improvements made to the revised manuscript, especially in highlighting the contributions, technical details, and novelty of the work. I believe it would still be beneficial to include recent state-of-the-art (SoTA) methods to demonstrate the advantages of the proposed approach. Additionally, based on Table 1 in the revised paper, without conducting a T-test, it is challenging to convincingly assert that the proposed method is superior, given the marginal performance differences of 0.1~0.2.

---

> > > ### Author Response · Authors · 2025-03-13
> > > **Reviewer Feedback Response: Appreciation and Additional Insights**
> > >
> > > Dear Reviewer, we sincerely appreciate your thoughtful feedback and your acknowledgment of the improvements made in our revised manuscript. Your insights are highly valuable in strengthening the clarity and rigor of our work.
> > >
> > > Regarding your suggestion to include recent state-of-the-art (SoTA) methods, we would like to highlight that we have provided a detailed comparison with the latest 3D self-supervised learning (SSL) SoTA models in Table 2 (Table 2: Performance analysis of 3D-SegSync with latest SOTA 3D SSL models for the MMWHS(CT) dataset). This table demonstrates the advantages of our proposed approach against recent advancements in the field.
> > >
> > > Additionally, we acknowledge your concern regarding the statistical significance of the performance differences. In our latest revision, we have included p-test scores in Table 2 to provide statistical validation for those results. However, due to time constraints, we were unable to conduct a similar statistical analysis for all dataset comparisons in Table 1. We appreciate your suggestion and will consider this in future work to further strengthen our claims.
> > >
> > > Once again, we sincerely thank you for your meticulous review and thoughtful recommendations. Your feedback has been immensely helpful, and we deeply appreciate your support in improving the quality of our work.

---

### Official Review · Reviewer_9bRQ · 2025-02-19

**Confidence:** 5
**Preliminary Rating:** 2
**Final Rating:** 4

**Summary:**

This study integrates DINOv2 with an xLSTM architecture for cardiac and brain image segmentation using CT and MRI images. Its main contributions include the pretraining step, the application of xLSTM, and an extensive evaluation supported by multiple diagrams and tables. However, there are some concerns regarding the contributions of the study.

**Strengths:**

1. The comparison results show improvements over state-of-the-art methods in both segmentation tasks. The authors have conducted an extensive evaluation using five metrics, two segmentation tasks, and multiple diagrams and tables.
2. The proposed framework utilizes recent techniques such as DINOv2 and xLSTM.

**Weaknesses:**

1. The Related works mentioned in the Introduction section are relatively old (2016-2021). The authors should include more recent work in the state-of-the-art for the SSL methods and the Heart and stroke lesion segmentation.
2. The proposed architecture is not thoroughly described. Details on the main components and the training procedure for both the SSL and xLSTM must be included to better present the methodological novelty, and the motivation and enhance reproducibility.
3. While the authors compared their proposed model with some state-of-the-art segmentation methods, they do not include Self-supervised learning approaches. They must have included other SSL methods, such as SwinUNETR[1], SwinMM[2], in order to present the superiority of their proposed SSL pretraining because one of their main contributions is the SSL pretraining. WIth the current comparisons they prove the superiority of using DINOv2 pretraining against not pretraining at all, but presenting comparisons with SSL would enhance their contribution.

[1] Y. Tang et al., “Self-Supervised Pre-Training of Swin Transformers for 3D Medical Image Analysis,” in 2022 IEEE/CVF Conference on Computer Vision and Pattern Recognition (CVPR), Jun. 2022, pp. 20 698–20 708, iSSN: 2575-7075.

[2] Y. Wang et al., “SwinMM: Masked Multi-view with Swin Transformers for 3D Medical Image Segmentation,” in Medical Image Computing and Computer Assisted Intervention – MICCAI 2023: 26th International Conference, Vancouver, BC, Canada, October 8–12,
2023, Proceedings, Part III. Berlin, Heidelberg: Springer-Verlag, Oct. 2023, pp. 486–496.

**Detailed Comments:**

1. In the tables, using a bold font for the best values in each metric will enhance the presentation.
2. Some details on the methodology should be included in the main manuscript rather than in the Appendix to better present the technical aspects of the approach.

**Justification Of The Final Rating:**

I would like to thank the authors for their detailed responses and revisions. The manuscript now addresses most of the key points raised in my earlier review, methodological details have been expanded, and the inclusion of a table comparing SSL methods on one dataset is a notable improvement. Additionally, the Related Work section is more comprehensive.

The primary remaining concern involves the novelty of combining DINOv2 with xLSTM. While the authors claim to adapt DINOv2 for three-dimensional data through xLSTM, the manuscript does not sufficiently clarify how this integration extends beyond a straightforward application of a 3D backbone model with 3D transformations. Despite this remaining question, the paper demonstrates merit, and I therefore recommend a Weak Accept.

**Justification Of The Preliminary Rating:**

There are some concerns about the contributions of the study, due to the already existing DINOv2 pretraining step and the application of the xLSTM. Additionally, the methodology of the study should be presented in more detail to better prove the motivation and contributions of the work.
Another major concern is the evaluation, which focuses solely on state-of-the-art segmentation methods without comparing the proposed method against state-of-the-art self-supervised learning (SSL) approaches. Given that DINOv2 is a key component of the study, the lack of such comparisons weakens the evaluation.

**Questions To Address In The Rebuttal:**

1. What is the motivation behind the main components of the architecture? Please include details on the methodology/architecture and implementation to enhance reproducibility.
2. In this regard, the novelty and contributions of the study must be clarified. DINOv2 is an already existing SSL method with wide applications and xLSTM is a newer architecture but studies have begun to appear.
3. Additional comparisons with state-of-the-art SSL methods (as reported in the Weaknesses) would support the superiority of the method.

---

> ### Author Response · Authors · 2025-03-07
> **Clarifications on Methodology, Novelty, and Comparisons with SOTA SSL Methods**
>
> Q1: The Related works mentioned in the Introduction section are relatively old (2016-2021). The authors should include more recent work in the state-of-the-art for the SSL methods and the Heart and stroke lesion segmentation.
> R1: Thank you for sharing your feedback. We have revised the related work
>
> Q2: The proposed architecture is not thoroughly described. Details on the main components and the training procedure for both the SSL and xLSTM must be included to better present the methodological novelty, and the motivation and enhance reproducibility.
> Answer: Thank you for your valuable feedback. To improve the clarity, novelty, and reproducibility of our approach, we have provided a thorough description of the proposed architecture, including both the SSL pretraining and the xLSTM components, along with the detailed training procedure. We have now provided detailed specifications, including the exact architecture of the xLSTM module (such as the number of layers, and hidden units), hyperparameters for both SSL pretraining and fine-tuning phases (e.g., learning rates, batch sizes, optimizer settings), and the data augmentation strategies and regularization methods employed during training in the Proposed ‘Method’ section. This will ensure that others can replicate our experiments and validate our findings effectively.
>
> Q3: While the authors compared their proposed model with some state-of-the-art segmentation methods, they do not include Self-supervised learning approaches. They must have included other SSL methods, such as SwinUNETR[1], SwinMM[2], in order to present the superiority of their proposed SSL pretraining because one of their main contributions is the SSL pretraining. WIth the current comparisons they prove the superiority of using DINOv2 pretraining against not pretraining at all, but presenting comparisons with SSL would enhance their contribution.
> R3: Thank you for your valuable feedback. We have now compared our proposed model with recent state-of-the-art self-supervised learning (SSL) approaches specifically designed for medical imaging, including Hi-End-MAE, SwinSSL, SimMIM, and Voco. Due to the limitation of time required for training all datasets on newly added 3D SSL SOTA models, we have presented this comparison on only the MMWHS(CT) whole-heart dataset (See Table 2 in the manuscript), to benchmark our SSL-based approach against these leading SSL methods.
>
> Q4: In the tables, using a bold font for the best values in each metric will enhance the presentation.
> R4: Thank you for the valuable suggestion. We have made the font bold for the best values in the tables now.
>
> Q5: Some details on the methodology should be included in the main manuscript rather than in the Appendix to better present the technical aspects of the approach.
> R5: Thank you for your suggestion. We have moved some useful information from the appendix to the main manuscript. We have added a new framework figure (Figure 1) to better explain the proposed model architecture.
>
> Concerns:
> What is the motivation behind the main components of the architecture? Please include details on the methodology/architecture and implementation to enhance reproducibility. 2. In this regard, the novelty and contributions of the study must be clarified. DINOv2 is an already existing SSL method with wide applications and xLSTM is a newer architecture but studies have begun to appear. 3. Additional comparisons with state-of-the-art SSL methods (as reported in the Weaknesses) would support the superiority of the method.
>
> Response: 1. Our architecture combines 3D self-supervised learning (SSL) with xLSTM to tackle challenges in large-scale 3D medical image segmentation, especially with limited labeled data. While DINOv2 uses Vision Transformers (ViTs) for 2D, we adapt its student-teacher mechanism for 3D data by integrating xLSTM to capture slice-to-slice dependencies in CT and MRI scans, improving segmentation accuracy. Detailed architecture and training specifications have been updated in the manuscript for reproducibility. 2. Thank you for your insightful feedback. We added the novelty and contribution of our work to the manuscript. 3. We agree that comparisons with additional 3D SSL methods, such as SwinUNETR and SwinMM, would strengthen our results. While we’ve compared our model with several baselines, these additional comparisons further demonstrate the effectiveness of our hybrid SSL-xLSTM approach for 3D medical segmentation (see Table 2).
>
> We sincerely thank the reviewers for their constructive feedback, which has greatly improved our manuscript. Their suggestions on clarifying technical innovations, expanding comparisons with SOTA methods, and providing more details on architecture and training procedures have been invaluable. We believe these revisions have strengthened the clarity and impact of our work. Thank you again for your thoughtful review.

---

### Official Review · Reviewer_BZkY · 2025-02-21

**Confidence:** 4
**Preliminary Rating:** 2
**Recommendation:** Poster
**Final Rating:** 4

**Summary:**

Authors propose a self-supervised learning framework for image segmentation in cardiac CT and brain MRI data. The framework is built upon DINOv2 student-teacher training and a 3D Vision-LSTM architecture model. The model is pre-trained on diverse datasets (cardiac: ImageCAS, ImageTBAD, TotalSegmentator, MMWHS, WHS++; brain: ISLES datasets, ATLAS, TBI). The model is evaluated on whole heart segmentation tasks and brain lesions segmentation tasks, against different architecture baselines.

**Strengths:**

- While some design choices are not fully justified, the method aligns with the broader goal of reducing reliance on labeled data and improving feature extraction in volumetric medical images.
- The model is evaluated on both CT and MRI datasets for whole-heart segmentation, stroke lesion segmentation, and traumatic brain injury (TBI) analysis. This demonstrates its potential for cross-modality and cross-application generalization.
- Authors provide code, promoting transparency and reproducibility.

**Weaknesses:**

- The choice of DINOv2 over other SSL methods is not well motivated—the authors should explain why this specific student-teacher approach was chosen instead of alternative self-supervised learning paradigms (e.g., masked autoencoders, VICReg, SimMIM). More critically, the xLSTM-UNet architecture is justified as a means to capture spatiotemporal dependencies, yet no temporal component is explicitly described or leveraged in the method. If the "temporal" aspect refers only to the sequential feature propagation within LSTMs, this should be clarified, as there is no apparent temporal dimension in static medical images like MRI and CT scans.
- The methodology and experimental settings lack sufficient clarity. It is unclear whether pre-training is performed simultaneously on both cardiac and brain imaging data. If this is the case, the benefits of such joint pretraining should be better justified, as cardiac and neurological structures are anatomically distinct. If not, the authors should clearly state that the proposed SSL framework is showcased for separate applications rather than as a universal model.
- The role of different datasets in pretraining, fine-tuning, and testing is not well explained. It is essential to provide a clear breakdown of how each dataset is used at each stage of training to ensure the evaluation process is meaningful and reproducible.
- While the authors highlight their SSL-based approach, they primarily compare it to different segmentation architectures (e.g., 3D-UNet, 3D-ResUNet, 3D-nnUNet), which appear to be trained in a supervised manner—but this is unclear in the text. A more appropriate comparison would be against state-of-the-art SSL methods for medical imaging. Evaluating against SSL-specific baselines would better demonstrate the relevance of the proposed pre-training framework.
- The presentation of results is difficult to follow and lacks clarity. Figures and tables should be formatted more effectively to improve readability. See detailed comments for suggestions on how to enhance clarity.

**Detailed Comments:**

Major comments:
- Introduction: The claim "By enhancing robustness, generalizability, and efficiency, our approach sets a new benchmark in medical image segmentation" is too broad and needs to be more specific. It is unclear whether all these aspects are fully demonstrated in the results. Additionally, the phrase "new benchmark in medical image segmentation" is overly generic and should be revised to specify in what context or task the method sets a benchmark.
- Authors appear to use CT exclusively for cardiac imaging and MRI exclusively for brain imaging. This raises a methodological question: what is the actual benefit of combining two different applications and two different modalities within one SSL framework? Does pretraining on both tasks improve performance, or would separate models be more effective?
- Validation datasets (MMWHS, WHS++) are used for pretraining, while their training sets are used for validation. What is the rationale behind this data split? This approach is unconventional and should be clearly justified.
- Section 2.2.3 starts with "In the third stage", but the previous stages are not clearly defined.
- How does the model perform without fine-tuning? Would the SSL pretraining alone produce useful representations? This is particularly relevant given the cross-modality setup—does pretraining on both cardiac and brain images actually contribute to better performance, or would separate pretraining pipelines be preferable?
- The evaluation only includes one segmentation task per modality (cardiac segmentation for CT, stroke lesion segmentation for MRI). This makes it unclear whether the proposed model truly achieves robustness and generalizability as claimed. Evaluating across multiple segmentation tasks per modality would support this claim more strongly.

Minor comments:
- Figure 1 caption lacks sufficient detail (this also applies to other figures and table captions). Captions should clearly explain the components of the framework for better readability.
- In the results figures, the proposed method is currently displayed in the second position. For clarity and emphasis, it would be better to place it either first or last to make comparisons easier.
- Two significant digits should be sufficient in tables. Additionally, standard deviations or statistical significance testing should be considered to support the reported improvements better.

**Justification Of The Final Rating:**

The authors have clearly addressed my concerns and provided substantial revisions that significantly improve the clarity of the methodology and evaluation. They have also included new experiments that strengthen the validity of their approach. The overall presentation is now much clearer and more structured, making the contributions more convincing.

Additionally, all reviewers seem to agree that the manuscript has improved considerably.

Given these enhancements, I am increasing my rating to 4: weak accept score.

**Justification Of The Preliminary Rating:**

While the paper presents an interesting self-supervised learning framework with promising segmentation results, the weaknesses outweigh the merits in its current form. The lack of clarity in methodology, dataset usage, and evaluation setup significantly hampers the ability to assess the contribution fully. The manuscript requires substantial improvements to enhance its claims' clarity, rigor, and validity.
Given these concerns, I recommend a preliminary rating of 2: Weak Reject.

**Questions To Address In The Rebuttal:**

- Address the concerns raised in the weaknesses section, particularly regarding the justification of claims, methodology clarity, and evaluation details.
- Improve the clarity of the manuscript by incorporating the points highlighted in the weaknesses and detailed comments, ensuring better explanation and presentation across the relevant sections.

---

> ### Author Response · Authors · 2025-03-07
> **Clarifications on Methodology, Dataset Usage, and Evaluation Setup**
>
> Q1: The choice of DINOv2 over other SSL methods is not well motivated. More critically, the xLSTM-UNet architecture is justified as a means to capture spatiotemporal dependencies
>
> Thank you for your valuable feedback and insightful comments. Specifically, regarding your concern about the choice of DINOv2 over other SSL methods, we have provided more details in the manuscript. We understand that the term 'temporal' could be misleading, as the images are static. Therefore, we have explicitly explained that the "temporal" aspect refers to the sequential feature propagation across adjacent slices in 3D CT/MRI scans, which is essential for capturing spatial dependencies across slices in 3D medical imaging.
>
> Q2: Unclear pre-training is performed simultaneously on both cardiac and brain imaging data.
>
> To clarify, pre-training was done separately for cardiac and brain datasets. We have added more details about this process in the manuscript. (see section ‘PROPOSED 3D SSL STUDENT-TEACHER MODEL’)
>
> Q3: The role of different datasets in pretraining, fine-tuning, and testing is not well explained.
>
> Thank you for your constructive comment. We apologize for the lack of clarity regarding the role of different datasets in pretraining, fine-tuning, and testing. We have now provided a more detailed breakdown of how each dataset was used at each stage of the training process in the section ‘Dataset’.
>
> Q4: A more appropriate comparison would be against state-of-the-art SSL methods for medical imaging.
>
> Thank you for your constructive comments and guidance. We have now compared our proposed model with recent SOTA SSL approaches specifically designed for 3D medical imaging (See Table 2 in the manuscript)
>
> Q5: Figures and tables should be formatted more effectively to improve readability.
>
> Thank you for your feedback. We have followed your detailed suggestions to refine the presentation, ensuring that results are conveyed in a more structured and accessible manner.
>
> Q6: Introduction: The claim is too broad and needs to be more specific.
>
> Thank you for your insightful feedback. We agree that the claim in the introduction was too broad and requires more specificity. We have now revised the statement to more precisely reflect the improvements demonstrated in our results.
>
> Q7: Authors appear to use CT exclusively for cardiac imaging and MRI exclusively for brain imaging.
>
> Thank you for your valuable input. To clarify, our SSL framework pre-trains separate models for cardiac and brain imaging, using CT/MRI for cardiac and MRI for brain imaging with large unlabeled datasets. The goal is to showcase the scalability and adaptability of our method across different anatomical structures and modalities. We recognize the potential benefits of a unified SSL model for multiple modalities and will explore this direction in future work.
>
> Q8: Validation datasets (MMWHS, WHS++) are used for pretraining, while their training sets are used for validation. What is the rationale behind this data split?
>
> The validation datasets (MMWHS, WHS++) lack ground truth labels as they are used for MICCAI challenge validation. For SSL pretraining, we utilized these unlabeled datasets, while for downstream segmentation, we used the labeled training sets. This clarification has been added to the manuscript (see section 'Dataset').
>
> Q9: Section 2.2.3 starts with "In the third stage", but the previous stages are not clearly defined
> Thank you for highlighting this, we have updated it now.
>
> Q10: How does the model perform without fine-tuning? Would the SSL pretraining alone produce useful representations? This is particularly relevant given the cross-modality setup
>
> As far as we understand, you are referring to the external evaluation where the model is fine-tuned on one dataset or modality and evaluated on the other. This is a very useful comparison that we have somehow missed at this stage, but we would strongly consider this aspect while further enhancing our approach for future applications.
>
> Q11: The evaluation only includes one segmentation task per modality (cardiac and brain). Evaluating across multiple segmentation tasks per modality would support this claim more strongly.
>
> We are grateful for your valuable input and observations. We used both CT and MRI for pretraining the SSL model, then fine-tuned it for cardiac segmentation on three datasets. 3D-SegSync, outperformed in both modalities, showing its adaptability and the benefits of multi-modal unlabeled data in self-supervised learning. In the future, we plan to test cross-modality performance by fine-tuning the model on one modality and testing on another.
>
> We acknowledge your concerns regarding the clarity of our methodology, dataset usage, and evaluation setup. To improve the manuscript, we’ve added a detailed figure on the methodology (Figure 1). We hope the revised manuscript effectively address your concerns. We sincerely appreciate the time and effort you dedicated to reviewing our manuscript

---

### Author Rebuttal · Authors · 2025-03-07

**Rebuttal:**

Regarding the concern about the choice of DINOv2 over other SSL methods, we have provided more details in the manuscript. We have explicitly explained the "temporal" aspect refers to the sequential feature propagation across adjacent slices in 3D CT/MRI scans.

Details about the pretraining strategy are updated. (see section: 2.2.1)

We have now provided a more detailed breakdown of how each dataset was used at each stage of the training process in the section ‘Dataset’.

More comparisons are given with 3D SOTA SSL models (See Table 2)

We have refined the presentation, ensuring that results are conveyed in a more structured and accessible manner.

We have now revised the ‘Introduction’ statement to more precisely reflect the improvements demonstrated in our results.

Our SSL framework pre-trains separate models for cardiac and brain imaging, using CT/MRI for cardiac and MRI for brain imaging with large unlabeled datasets. The goal is to showcase the scalability and adaptability of our method across different anatomical structures and modalities.

The MMWHS and WHS++ datasets lack ground truth labels as they are used for MICCAI challenge validation; we used these unlabeled datasets for SSL pretraining (see section: Dataset)

We used both CT and MRI for pretraining the SSL model, then fine-tuned it for cardiac segmentation on three datasets. 3D-SegSync, outperformed in both modalities, showing its adaptability and the benefits of multi-modal unlabeled data in self-supervised learning.

We acknowledge the concerns regarding the clarity of our methodology, dataset usage, and evaluation setup. To improve the manuscript, we’ve added a detailed figure on the methodology (Figure 1).

To improve the clarity, novelty, and reproducibility of our approach, we have provided a thorough description of the proposed architecture, including both the SSL pretraining and the xLSTM components, along with the detailed training procedure.

We have now compared our proposed model with recent state-of-the-art self-supervised learning (SSL) approaches specifically designed for medical imaging, including Hi-End-MAE, SwinSSL, SimMIM, and Voco. (See Table 2 in the manuscript)

We have moved some useful information from the appendix to the main manuscript (see Figure 1)

We have updated the manuscript to clarify the motivation behind adopting the DINOv2 student-teacher structure.

Supporting Material: Revised Manuscript, Detailed Reviewer Responses

**Supporting Material:**

/attachment/3e9e5f2d1d0a2ed5947ef559e21cfd83f49fb5e5.zip

---

### Meta-Review · Area_Chair_Usbi · 2025-03-21

**Recommendation:** Reject
**Confidence:** 4

**Metareview:**

Authors propose a self-supervised learning framework for image segmentation in cardiac CT and brain MRI data. The framework is built upon DINOv2 student-teacher training and a 3D Vision-LSTM architecture model. The model is pre-trained on diverse datasets (cardiac: ImageCAS, ImageTBAD, TotalSegmentator, MMWHS, WHS++; brain: ISLES datasets, ATLAS, TBI). The model is evaluated on whole heart segmentation tasks and brain lesions segmentation tasks, against different architecture baselines.
One remaining concern involves the novelty of combining DINOv2 with xLSTM. While the authors claim to adapt DINOv2 for three-dimensional data through xLSTM, the manuscript does not sufficiently clarify how this integration extends beyond a straightforward application of a 3D backbone model with 3D transformations.
Moreover there is a lack of a baseline.
The major concern is the lack of statistical analysis, which is required to validate the method. Only Average Dice and HD are given, the standard deviation is missing for most of the tables